# *CHiP*: CROSS-MODAL HIERARCHICAL DIRECT PREFERENCE OPTIMIZATION FOR MULTIMODAL LLMS

**Jinlan Fu**[1]   **Shenzhen Huangfu**[1,2*]   **Hao Fei**[1]   **Xiaoyu Shen**[3]
**Bryan Hooi**[1]   **Xipeng Qiu**[2†]   **See-Kiong Ng**[1]

[1]National University of Singapore    [2]Fudan University
[3]Digital Twin Institute, Eastern Institute of Technology, Ningbo

jinlanjonna@gmail.com, shenzhenhuangfu@gmail.com

## ABSTRACT

Multimodal Large Language Models (MLLMs) still struggle with hallucinations despite their impressive capabilities. Recent studies have attempted to mitigate this by applying Direct Preference Optimization (DPO) to multimodal scenarios using preference pairs from text-based responses. However, our analysis of representation distributions reveals that multimodal DPO struggles to align image and text representations and to distinguish between hallucinated and non-hallucinated descriptions. To address these challenges, In this work, we propose a **C**ross-modal **Hi**erarchical Direct **P**reference Optimization (CHiP) to address these limitations. We introduce a visual preference optimization module within the DPO framework, enabling MLLMs to learn from both textual and visual preferences simultaneously. Furthermore, we propose a hierarchical textual preference optimization module that allows the model to capture preferences at multiple granular levels, including response, segment, and token levels. We evaluate CHiP through both quantitative and qualitative analyses, with results across multiple benchmarks demonstrating its effectiveness in reducing hallucinations. On the Object Hal-Bench dataset, CHiP outperforms DPO in hallucination reduction, achieving improvements of 52.7% and 55.5% relative points based on the base model Muffin and LLaVA models, respectively. We make all our datasets and code publicly available. [1]

## 1 INTRODUCTION

The emergence of large language models (LLMs) has demonstrated unprecedented intelligence (Chiang et al., 2023; Touvron et al., 2023; AI@Meta, 2024), bringing us closer to achieving human-level AI. Concurrently, building on the foundation of text-based LLMs, research in Multimodal Large Language Models (MLLMs) has also rapidly surged, leading to the development of powerful multimodal models such as GPT-4V (OpenAI, 2023), BLIP2 (Li et al., 2023a), and LLaVA (Liu et al., 2024c). The current MLLMs typically integrate the visual encoder into the text-oriented backbone LLMs through a connector to achieve the understanding of visual signals (Liu et al., 2024c; Yao et al., 2024). Although MLLMs have achieved impressive results, hallucination remains a significant challenge, where the model's output is not based on the visual input (Bai et al., 2024; Jiang et al., 2024).

With the help of Direct Preference Optimization (DPO) (Rafailov et al., 2024) (Fig. 2-(a)), text-oriented LLMs have achieved satisfactory alignment with human preferences, which can help prevent hallucinations and enhance their ability to meet human needs. However, the alignment techniques for Multimodal LLMs (MLLMs) remain underexplored. A natural approach is to extend DPO from the text modality to multimodal contexts through multimodal DPO (Pi et al., 2024; Sarkar et al., 2024a) (Fig. 2-(b)). However, simply replacing text preference data with multimodal preference data

---

*Work done during an internship at NUS.

†Corresponding author.

[1]https://github.com/LVUGAI/CHiP

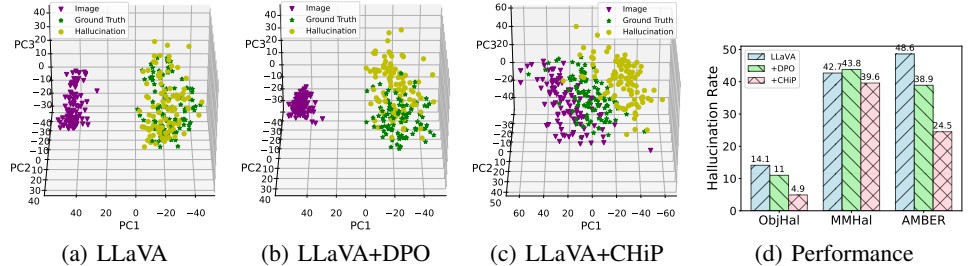

(a) LLaVA     (b) LLaVA+DPO     (c) LLaVA+CHiP     (d) Performance

Figure 1: Comparison of representation distributions and performance across models. Representations are constructed by selecting 150 samples (images, non-hallucinated descriptions, and hallucinated descriptions). The image or text semantics are represented using the last token embedding from the LLM. (d) is the hallucination rate (lower the better) comparison of different models on hallucination benchmarks, namely ObjHal, MMHal, and AMBER. Findings: *(1) DPO struggles to align image and description representations and to effectively distinguish between hallucinated and non-hallucinated descriptions. (2) The proposed CHiP method, which incorporates both image and fine-grained text preferences, achieves better alignment between images and ground-truth descriptions while increasing the distance between ground-truth and hallucinated descriptions. (3) CHiP outperforms DPO and original LLaVA in terms of hallucination rate.*

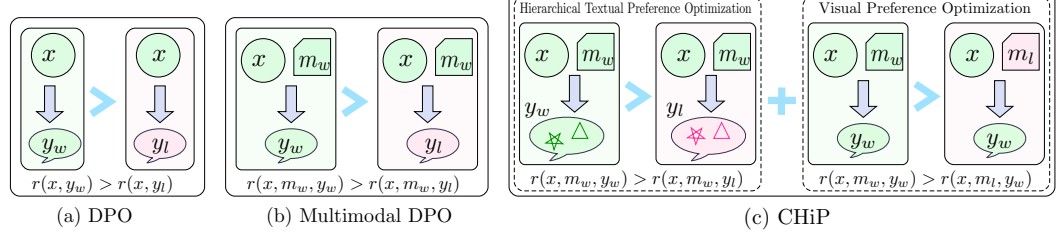

(a) DPO         (b) Multimodal DPO         (c) CHiP

Figure 2: Comparison of preference optimization in different scenarios: (a) DPO, (b) Multimodal DPO, and (c) Cross-modal Hierarchical Direct Preference Optimization (CHiP). $x$ represents the instruction. $y_w$ denotes the response preferred by the human over $y_l$. $m_w$ represents the image that is more likely to generate the preferred response $y_w$ than $m_l$. ★ (▲) and ☆ (△) represent the segments (tokens) involved in the hierarchy reward calculation in the preferred and unpreferred responses.

is insufficient to handle complex multimodal scenarios. In this work, we identify the limitations of multimodal DPO by visualizing the representation distributions of both images and texts and propose solutions to overcome these challenges. Ideally, for well-aligned MLLMs, the representations of an image and its ground-truth description should be as close as possible, while the representations of ground-truth and hallucinated descriptions should be more distant. Fig. 1 shows a visualization of the last token representations of image and text in the LLM (specifically LLaMA (Touvron et al., 2023)) for LLaVA-1.6 (Liu et al., 2024b), using 150 samples (image, ground-truth description, and hallucinated description). Fig. 1-(a, b, d) highlights the limitations of multimodal DPO, showing its difficulty in aligning image and description representations and distinguishing between hallucinated and non-hallucinated descriptions, hindering performance improvement in hallucination evaluations.

To address these limitations, we propose Cross-modal Hierarchical Direct Preference Optimization (CHiP) (Fig. 2-(c)), which enhances the alignment from multiple textual granularities (e.g., response, segment, token levels) and visual preferences. Specifically, we introduce *Visual Preference Optimization* by constructing visual preference pairs, allowing the model to learn preferences from both text and visual modalities, aligning text and image representations more closely. Moreover, we introduce a *Hierarchical Textual Preference Optimization* to allow MLLMs to acquire preference information at multiple granular levels, namely, response, segment, and token, enhancing their ability to differentiate between hallucinated and non-hallucinated text. To validate the efficacy of CHiP, we evaluate it on four popular hallucination benchmarks under LLaVA-1.6 and Muffin frameworks. The results show that CHiP outperforms GPT-4V significantly on the evaluated benchmarks. Moreover, on the Object HalBench dataset, based on Muffin and LLaVA-1.6 models, CHiP outperforms DPO in hallucination reduction, with performance improvements of 52.7% and 55.5% relative points, respectively.

To sum up, our contributions are threefold:

- We analyze the limitations of multimodal DPO through image and text representation distributions, emphasizing its failure to achieve cross-modal semantic alignment and distinguish between hallucinated and non-hallucinated descriptions.
- We propose CHiP to address these limitations. CHiP includes a hierarchical textual preference optimization module to capture fine-grained (i.e., response, segment, and token) preferences and a visual preference optimization module to extract cross-modal preferences.
- We equipped CHiP with various MLLMs, and the results of multiple datasets demonstrate that CHiP reduces hallucinations and enhances cross-modal semantic alignment.

## 2 RELATED WORK

Multimodal Large Language Models (MLLMs) (Liu et al., 2024c; Bai et al., 2023; Dai et al., 2023; He et al., 2024) play a crucial role in multimodal understanding and reasoning tasks by processing both images and text. Their development has been fueled by progress in open-source LLMs (Touvron et al., 2023; AI@Meta, 2024; Chiang et al., 2023) and cutting-edge image encoders (Radford et al., 2021b; Wang et al., 2023c; Li et al., 2022).

However, misalignment between images and text causes MLLMs to face issues like hallucinations and errors. Mitigating hallucinations is a crucial research of MLLMs. Hallucination mitigation strategies generally fall into two categories: training-free and training-based methods. Training-free approaches (Huang et al., 2024; Yin et al., 2023) handle potential hallucinations by post-processing MLLMs' outputs. On the other hand, training-based approaches aim to reduce hallucinations through instruction fine-tuning (Liu et al., 2023b; Zhang et al., 2024) or preference learning (Gunjal et al., 2024; Li et al., 2023b; Sun et al., 2023; Yu et al., 2024a; Deng et al., 2024; Wang et al., 2024). For example, REVERIE (Zhang et al., 2024) is a reflective instruction tuning method that incorporates rationale learning into visual instruction tuning. As for the preference learning, for example, Gunjal et al. (2024) proposed Fine-grained DPO (FDPO) and trained a fine-grained multimodal reward model based on InstructBLIP (Dai et al., 2023).

Different from previous research, we address the visual-language preference misalignment in MLLMs by introducing a novel cross-modal hierarchical DPO (i.e., CHiP), which simultaneously optimizes preferences in both the text and image modalities from a fine-grained perspective. Our approach demonstrates better alignment between the two modalities from the visualization of representation and reduced hallucination generation.

## 3 PRELIMINARIES

Direct Preference Optimization (DPO) (Rafailov et al., 2024) is primarily a preference optimization method that focuses on aligning language models with human preferences without the need for explicit reward modeling or reinforcement learning. Given a model to be optimized $\pi_\theta$, and the reference policy $\pi_{\text{ref}}$, which is a supervised fine-tuning model, the RL optimization of RLHF can be formulated as:

$$\max_{\pi_\theta} \mathbb{E}_{x \sim \mathcal{D}, y \sim \pi_\theta(y|x)} \big[ r(x, y) \big] - \beta \mathbb{D}_{\text{KL}} \big[ \pi_\theta(y \mid x) \,||\, \pi_{\text{ref}}(y \mid x) \big] . \tag{1}$$

By maximizing the KL-constrained reward objective to obtain the optimal solution and establishing a mapping between the reward model and the optimal policy, the representation of the reward function is derived as follows:

$$r(x, y) = \beta \log \frac{\pi_\theta(y|x)}{\pi_{\text{ref}}(y|x)} + \beta \log Z(x) , \tag{2}$$

where $x$ is the input instruction, $y$ is the response, $\beta$ is a constant, and $Z(x)$ is the partition function.

Given the chosen response $y_w$, where the evaluator preferred it over the rejected response $y_l$, DPO is expected to learn to maximize the reward difference between chosen ($y_w$) and rejected responses ($y_l$). The preference optimization objective becomes:

$$\begin{aligned} \mathcal{L}_{\mathcal{DPO}} &= -\mathbb{E}_{(x, y_w, y_l)} \big[ \log \sigma(r(x, y_w) - r(x, y_l)) \big] \\ &= -\mathbb{E}_{(x, y_w, y_l)} \big[ \log \sigma(\beta \log \frac{\pi_\theta(y_w|x)}{\pi_{\text{ref}}(y_w|x)} - \beta \log \frac{\pi_\theta(y_l|x)}{\pi_{\text{ref}}(y_l|x)} ) \big] , \end{aligned} \tag{3}$$

where DPO learns preferences based on the ranking of the entire response, and the action score can be formulated as:

$$\log \pi(y|x) = \sum_{y_i \in y} \log p(y_i|x, y_{<i}),$$ (4)

where $y_i$ denotes the $i$-th token of the response $y$. During DPO training, the reference model $\pi_{\text{ref}}(y|x)$ is usually kept fixed while the policy model $\pi_\theta(y|x)$ is updated.

# 4 METHODOLOGY: CROSS-MODAL HIERARCHICAL DIRECT PREFERENCE OPTIMIZATION

In this section, we will introduce the Cross-modal Hierarchical Direct Preference Optimization (CHiP). CHiP consists of two modules: (1) Hierarchical Textual Preference Optimization, which incorporates preference optimization at the response, segment, and token levels; and (2) Visual Preference Optimization, which addresses the overlooked visual information.

## 4.1 HIERARCHICAL TEXTUAL PREFERENCE OPTIMIZATION

Image-based responses are often long and complex, and response-level preference optimization relies on rough rankings of response quality without clearly identifying which segments or tokens contain hallucinations. This makes it challenging to assign credit to desirable behaviors, leading to reward hacking (Laidlaw et al., 2024) and the need for a large amount of labeled data. Therefore, we introduce the Hierarchical Textual Preference Optimization module to assign rewards from fine-grained.

For MLLMs, each sample includes an image ($m$) in addition to prompt $x$, chosen response $y_w$, and rejected response $y_l$. Multimodal DPO relies on both prompt $x$ and image $m$ to select the preferred response from $\{y_w, y_l\}$. Next, we will provide a detailed illustration of the three levels of preference optimization for text: response-level, segment-level, and token-level.

**Response-level Preference Optimization ($\mathcal{L}_{\mathcal{DPO}r}$).** At the response level, DPO in the MLLMs' scenario aims to maximize $\sigma(r(x, m_w, y_w) - r(x, m_w, y_l))$, and the objective function can be formulated as:

$$\mathcal{L}_{\mathcal{DPO}r} = -\log \sigma \left( \beta \log \frac{\pi_\theta(y_w|m, x)}{\pi_{\text{ref}}(y_w|m, x)} - \beta \log \frac{\pi_\theta(y_l|m, x)}{\pi_{\text{ref}}(y_l|m, x)} \right),$$ (5)

$\log \pi(y|x)$ can be formulated as:

$$\log \pi(y|x, m) = \sum_{y_i \in y} \log p(y_i|x, m, y_{<i}),$$ (6)

where $y_i$ denotes the $i$-th token of the response $y$.

**Segment-level Preference Optimization ($\mathcal{L}_{\mathcal{DPO}s}$).** Intuitively, the corrected segments, particularly entity nouns, play a crucial role in eliminating hallucinations and should be assigned more rewards. Following Yu et al. (2024a), we assign higher rewards to the segments that differ between the chosen response and the rejection response. Based on Eq. 6, the action score for the segment-level can be denoted as:

$$\log \pi^{\text{seg}}(y|x, m) = \frac{1}{C} \left( \sum_{y_i \in y} \log p(y_i|x, m, y_{<i}) + \gamma \sum_{y_i \in y_c} \log p(y_i|x, m, y_{<i}) \right),$$ (7)

where $y_c$ indicates the segments where changes have occurred. $y_i$ denotes the $i$-th token of the response $y$. To prevent the model from being misled into giving higher scores to longer responses, $\frac{1}{C}$ is used as a normalization factor, where $C = |y| + \gamma * |y_c|$. By substituting Eq. 7 into Eq. 5, we can obtain the objective function of segment-level preference optimization $\mathcal{L}_{\mathcal{DPO}s}$.

**Token-level Preference Optimization ($\mathcal{L}_{\mathcal{PO}k}$).** For most previous methods, the optimization objective of DPO was constructed based on sentence-level KL divergence Eq. 1). However, the output generated from images is an autoregressive sequence, so aligning MLLMs with human values at the

token level is natural. Finer-grained alignment not only improves alignment performance but also helps the model maintain diversity (Zeng et al., 2024). Unlike response-level optimization, which computes a single reward and KL divergence for the entire response, token-level optimization evaluates each token individually, with the cumulative token values forming the score of response. The sequential KL divergence can be defined as:

$$\mathcal{L}_{\mathcal{PO}k} = sg\left(\beta D_{\text{SeqKL}}\left(x, m, y_w; \pi_{\text{ref}} \| \pi_\theta\right)\right) - \beta D_{\text{SeqKL}}\left(x, m, y_l; \pi_{\text{ref}} \| \pi_\theta\right), \tag{8}$$

where $sg$ represents the stop-gradient operator, and

$$D_{\text{SeqKL}}(x, m, y; \pi_{\text{ref}} \| \pi_\theta) = \sum_{t=1}^{T} D_{\text{KL}}(\pi_{\text{ref}}(y|x, y^{<t}) \| \pi_\theta(y|x, y^{<t})). \tag{9}$$

**Hierarchical Textual Preference Optimization (HDPO).** includes the response, segment, and token-level preference optimization. It can be formulated as:

$$\mathcal{L}_{\mathcal{HDPO}} = \mathcal{L}_{\mathcal{DPO}r} + \lambda \mathcal{L}_{\mathcal{DPO}s} + \gamma \mathcal{L}_{\mathcal{PO}k}, \tag{10}$$

where $\lambda$ and $\gamma$ represent the weights of $\mathcal{L}_{\mathcal{DPO}r}$ and $\mathcal{L}_{\mathcal{PO}k}$, respectively.

## 4.2 Visual Preference Optimization

To mitigate MLLMs' over-reliance on large language models, we next introduce our Visual Preference Optimization Module. This module forces the model to make preference judgments based on visual information by constructing pairs of images with preferences as variables. Given a pair of images $(m_w, m_l)$, where $m_w$ allows the prompt $x$ to better match the chosen response $y_w$ compared to $m_l$, Visual Preference Optimization tries to maximize $\sigma(r(x, m_w, y_w) - r(x, m_l, y_w))$, and the objective function can be formulated as:

$$\mathcal{L}_{\mathcal{DPO}v} = -\log \sigma \left(\beta \log \frac{\pi_\theta(y_w|m_w, x)}{\pi_{\text{ref}}(y_w|m_w, x)} - \beta \log \frac{\pi_\theta(y_w|m_l, x)}{\pi_{\text{ref}}(y_w|m_l, x)}\right), \tag{11}$$

where the rejection image $m_l$ can be generated by rotating, cropping, or adding noise to the chosen image $m_w$. The objective of CHiP is a combination of the hierarchical textual (Eq. 10) and visual (Eq. 11) preference optimizations:

$$\mathcal{L}_{\mathcal{CHiP}} = \mathcal{L}_{\mathcal{DPO}v} + \mathcal{L}_{\mathcal{DPO}r} + \lambda \mathcal{L}_{\mathcal{DPO}s} + \gamma \mathcal{L}_{\mathcal{PO}k}. \tag{12}$$

Since the entire response and image encapsulate modality semantics, we assign a weight of 1 (fully consider) to response-level ($\mathcal{L}_{\mathcal{DPO}r}$) and visual preference optimization ($\mathcal{L}_{\mathcal{DPO}v}$). $\lambda$ and $\gamma$ ($< 1$) (partially consider) adjust the contributions of segment- and token-level preference optimizations. We refer to the model that only considers response-level and visual preference optimization as Cross-modal Direct Preference Optimization (CMDPO), which can be formulated as: $\mathcal{L}_{\mathcal{CMDPO}} = \mathcal{L}_{\mathcal{DPO}v} + \mathcal{L}_{\mathcal{DPO}r}$.

Hierarchical textual preference optimization module tries to maximizes $\sigma(r(x, m, y_w) - r(x, m, y_l))$ from different levels, while the visual preference optimization module tries to maximizes $\sigma(r(x, m_w, y_w) - r(x, m_l, y_w))$. The combination of the them allows MLLMs to choose preferences based on both fine-grained textual and visual modalities.

## 5 Experiment and Results

In this section, we empirically investigate the effectiveness of CHiP in reducing hallucination.

### 5.1 Experimental Settings

**Comparing Models.** We consider applying CHiP to two different multimodal LLMs: LLaVA-1.6 (Liu et al., 2024b) and Muffin (Yu et al., 2023). For LLaVA-1.6, we have chosen the model size of 7B, using CLIP (Radford et al., 2021a) as the visual encoder and Vicuna-1.5-7B (Zheng et al., 2023) as the LLM backbone. For Muffin, we have chosen a model size of 13B, using BEiT3 (Wang et al., 2023b) as the visual module and 13B Vicuna v1.0 (Chiang et al., 2023) as the LLM backbone, and a version fine-tuned on the VQAv2 dataset (Goyal et al., 2017) (released by Yu et al. (2024a)).

**Training Data.** There are several publicly available training datasets that include preference pairs for multimodal hallucinations. Here, we choose to use the RLHF-V-Dataset (Yu et al., 2024a;b) with 5k training samples as our training dataset.

**Baselines.** We primarily compare CHiP with standard DPO based on the same models. While other multimodal LLMs cannot be directly compared due to differences in base models, preference data, and alignment methods, we provide these results for reference. LLaVA (Liu et al., 2024c), Muffin (Yu et al., 2023), LRV (Liu et al., 2023a), LLaVA-RLHF (Sun et al., 2023), InstructBLIP (Dai et al., 2023), Qwen-VL-Chat (Bai et al., 2023), LLaVA 1.5 (Liu et al., 2023c), RLHF-V (Yu et al., 2024a), HALVA (13B) (Sarkar et al., 2024b).

**Benchmarks and Evaluation Metrics.**

- **Object HalBench (ObjHal)** (Rohrbach et al., 2018) is a widely used benchmark for evaluating object hallucination. To improve evaluation stability, the benchmark includes 8 diverse prompts and is tested on 300 instances. **Metrics**: Following Yu et al. (2024a); Wang et al. (2024), we report both the *response-level hallucination rate* (`R.`) and *mention-level hallucination rate* (`M.`).
- **MMHal-Bench (MMHal)** (Sun et al., 2023) is a question-answering benchmark that covers 8 question categories and 12 object topics. **Metrics:** It uses GPT-4 to assess response quality (`Ova.`) and hallucination rates (`R.`).
- **HallusionBench** (Guan et al., 2024) evaluates visual illusions and knowledge hallucinations, featuring 346 images and 1129 questions. It was the GPT4-assisted evaluation. **Metrics:** Question Pair Accuracy (`qA`), Figure Accuracy (`fA`), and All Accuracy (`aA`).
- **AMBER** (Wang et al., 2023a) was designed to be evaluated without LLM assistance. Following previous works (Wang et al., 2024), we only consider the generative tasks. **Metrics:** (a) CHAIR (Rohrbach et al., 2018) (`CHAIR`); (b) Object coverage of responses (`Cover`); (c) Response-level hallucination (`Hal`); (d) Human cognition hallucination (`Cog`).

**Implementation Details.** We train the Muffin (13B) (Yu et al., 2023) and LLaVA-1.6 (7B) (Liu et al., 2024b) with CHiP for 3 epochs, with learning rate of 5e-7 and a batch size of 32. For the training time, LLAVA-1.6 took about three hour to train with CHiP on 4 H100 GPUs, while Muffin took approximately five hours. Hyperparameter: Since our training dataset is RLHF-V dataset Yu et al. (2024a), we followed Yu et al. (2024a) to set the hyperparameter $\beta = 0.5$ and followed (Zeng et al., 2024) to set $\gamma = 0.1$ for token-level preference optimization. As for the weight of segment-level preference optimization, namely $\lambda$, we set to $\lambda = 1$ and $\lambda = 3$ for the Muffin and LLava dataset set (Sec. 5.3). *How to Identify Hallucinated Segments?* Our training dataset, RLHF-V contains both pre-correction (hallucinated) and post-correction (non-hallucinated) descriptions. We enumerate all segments longer than two tokens in rejected responses and classify those absent in accepted responses as hallucinations. *How to construct the rejected images?* The rejected images are built based on the chosen image of the forward diffusion process at $T = 500$ steps (Sec. 6.2).

## 5.2 RESULTS AND OBSERVATIONS

Tab. 1 presents the experimental results of applying CHiP to the LLaVA-1.6 and Muffin on four popular hallucination benchmarks. The main findings are listed below: (1) *CHiP significantly reduces hallucinations of base models Muffin and LLaVA-1.6.* Compared to the base model Muffin (LLaVA), CHiP reduced response- (`R.`) and mention-levels (`M.`) hallucinations by 71.2% (65.3%) and 66.4% (56.7%) relative point on the ObjHal dataset and human cognitive hallucinations (`Cog`) by 57.1% (61.9%) and Hal by 45.2% (49.6%) relative point on the AMBER dataset. Furthermore, the consistent improvements of CHiP in question pairs (`qA`) and visual understanding types (`fA`) on the HallucinationBench, as well as in overall object hallucinations (`Overall`) on the MMHal dataset. (2) *Based on Muffin and LLaVA, CHiP consistently outperforms DPO in reducing hallucination on the four benchmarks.* This indicates that CHiP, which includes the visual preference optimization module and the hierarchical textual preference optimization module, can effectively improve preference alignment performance. (3) *LLaVA and Muffin with CHiP achieve fewer hallucinations compared to GPT-4 on the ObjHal and AMBER datasets.* Compared to GPT-4, LLaVA (Muffin) with CHiP reduced hallucination rates at the response level and mention level by 64.0% (54.4%) and 56.2% (46.6%) relative point respectively on the Object HalBench. On the AMBER dataset, the hallucination rate for the `Cog` metric decreased by 42.3% (38.5%) relative point, with continuous reductions observed across several other categories as well.

Table 1: The results of hallucination evaluation on the Object HalBench (ObjHal), MMHal-Bench (MMHal), HallusionBench, and AMBER datasets. Values in **bold** indicate the best performance under the same setting. "↑" indicates that a higher value is better for this metric, while "↓" indicates that a lower value is better. The baseline results are reported in (Yu et al., 2024a) for ObjHal and MMHal, in (Guan et al., 2024) for HallucinationBench, and in (Wang et al., 2024) for AMBER.

| Model | ObjHal | | MMHal | | HallusionBench | | | AMBER | | | |
|---|---|---|---|---|---|---|---|---|---|---|---|
| | R.↓ | M.↓ | Ova.↑ | R.↓ | qA↑ | fA↑ | aA↑ | CHAIR↓ | Cover↑ | Hal↓ | Cog↓ |
| *Referenced Results (Not Directly Comparable)* | | | | | | | | | | | |
| LLaVA-1.0 (Liu et al., 2024c) | 63.0 | 29.5 | - | 70.8 | - | - | - | - | - | - | - |
| Muffin (Yu et al., 2023) | 50.5 | 24.5 | - | 68.8 | - | - | - | - | - | - | - |
| LRV (Liu et al., 2023a) | 32.3 | 22.3 | - | 78.1 | 8.8 | 13.0 | 42.8 | - | - | - | - |
| LLaVA-RLHF (Sun et al., 2023) | 38.1 | 18.9 | 2.5 | 57.0 | - | - | - | 7.7 | 52.1 | 39.0 | 4.4 |
| InstructBLIP (Dai et al., 2023) | 25.9 | 14.3 | 2.1 | 58.0 | 9.5 | 10.1 | 45.3 | 8.8 | 52.2 | 38.2 | 4.4 |
| Qwen-VL-Chat (Bai et al., 2023) | 43.8 | 20.0 | 2.9 | 43.0 | 5.9 | 6.7 | 39.2 | 6.6 | 53.2 | 31.0 | 2.9 |
| LLaVA-1.5 (Liu et al., 2023c) | 46.3 | 22.6 | 2.4 | 52.1 | 10.6 | 24.9 | 46.9 | 7.8 | 51.0 | 36.4 | 4.2 |
| RLHF-V (Yu et al., 2024a) | 12.2 | 7.5 | 2.5 | 51.0 | - | - | - | 6.3 | 46.1 | 25.1 | 2.1 |
| HALVA (Sarkar et al., 2024b) | - | - | - | - | 13.9 | 20.1 | 49.1 | - | - | - | - |
| GPT-4V (OpenAI, 2023) | 13.6 | 7.3 | - | 31.3 | 28.8 | 39.9 | 65.3 | 4.6 | 67.1 | 30.7 | 2.6 |
| Muffin (13B) | 21.5 | 11.6 | 2.4 | 60.42 | 16.0 | 20.8 | 50.9 | 8.0 | **48.3** | 32.1 | 3.5 |
| +DPO | 13.1 | 6.6 | 2.5 | 52.1 | 17.4 | 23.4 | 52.5 | 6.2 | 46.9 | 26.5 | 2.5 |
| +CHiP | **6.2** | **3.9** | **2.6** | **49.0** | **19.1** | **24.9** | **54.0** | **4.4** | 45.3 | **17.6** | **1.5** |
| LLaVA-1.6 (7B) | 14.1 | 7.4 | 2.8 | 42.7 | 15.8 | 20.8 | 51.6 | 8.3 | **61.0** | 48.6 | 4.2 |
| +DPO | 11.0 | 6.6 | 2.7 | 43.8 | 22.2 | **28.3** | 56.6 | 5.9 | **61.0** | 38.9 | 3.0 |
| +CHiP | **4.9** | **3.2** | **2.9** | **39.6** | **23.5** | 26.0 | **58.5** | **3.7** | 57.8 | **24.5** | **1.6** |

## 5.3 ABLATION STUDY

**Effect of Component Combination.** To evaluate the contribution of each component in CHiP and the effect of their combinations, we conducted a comprehensive ablation study on CHiP based on LLaVA. The experimental results are shown in Tab. 2. The main observations are as follows: (1) *Both hierarchical textual preference optimization (HDPO) and visual preference optimization (CMDPO) are effect.* On the ObjHal and MMHal datasets, both HDPO (CHiP-$\mathcal{L}_{\mathcal{DPO}_v}$) and CMDPO ($\mathcal{L}_{\mathcal{DPO}_s}$-$\mathcal{L}_{\mathcal{PO}_t}$) outperform DPO. This suggests that: (a) more granular preference signals can reduce label ambiguity at the response level (DPO), helping the model learn more effectively; (b) The introduction of visual preference optimization enhances the model's alignment between the image and text. (2) *The combination of visual preference optimization and hierarchical preference optimization strategies makes DPO the most powerful.* On both evaluation datasets, CMDPO introduced segment-level ($\mathcal{L}_{\mathcal{DPO}_s}$), token-level ($\mathcal{L}_{\mathcal{PO}_t}$), or both segment- and token-levels (CHiP), resulting in a consistent and significant reduction in hallucination rates across different evaluation perspectives, such as response level and mention level. Specifically, when CMDPO incorporates both segment-level and token-level optimization (CHiP), the response- and mention-level hallucination rate on ObjHal datasets decreased by 49.6% and 41.3%, respectively.

**Human Evaluation.** Due to incomplete text annotations on the MMHal, GPT-4 couldn't reliably detect hallucinations. To make the results more reliable, we invited experts to manually annotate the data to compare CHiP with DPO based on LLaVA. As shown in Figure 3, CHiP and DPO performed equally on 63.5% of samples, with CHiP winning 24%. In the 36.5% of samples where a distinction was possible, CHiP outperformed DPO in 31.6%.

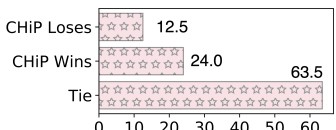

Figure 3: Human evaluation results on MMHal-Bench (MMHal).

**Strength of Hierarchical Textual Preference Optimization.** Hierarchical text preference optimization includes preference optimization at the response, segment, and token levels. Here, we discuss the impact of their weights. We fully consider response-level since its global textual semantics by setting its parameter to 1 (Eq. 12). and following Touvron et al. (2023), we set the

token-level weight $\gamma = 0.1$. As for segment-level, given its crucial role in providing fine-grained feedback on the preference of hallucinations, we fully explore the range of its weight $\lambda$ (as shown in Eq. 10). From the results of Fig. 4, we found that the best performance was achieved when $\lambda = 1$ and $\lambda = 3$ for the Muffin and LLaVA frameworks, respectively, and we adopted these settings in all the experiments presented in this paper.

Table 2: The ablation results of CHiP based on LLaVA. Values in **bold** denote the best performance.

| Model | ObjHal | | MMHal | |
|---|---|---|---|---|
| | R.↓ | M.↓ | Ova.↑ | R.↓ |
| DPO | 11.03 | 6.61 | 2.73 | 43.75 |
| CHiP | **4.92** | **3.21** | **2.89** | **39.63** |
| $-\mathcal{L}_{\mathcal{DPO}v}$ | 9.19 | 5.77 | 2.70 | 42.40 |
| $-\mathcal{L}_{\mathcal{DPO}s}$ | 8.55 | 5.16 | 2.69 | 40.63 |
| $-\mathcal{L}_{\mathcal{PO}t}$ | 6.08 | 3.77 | 2.71 | 40.75 |
| $-\mathcal{L}_{\mathcal{DPO}s}$-$\mathcal{L}_{\mathcal{PO}t}$ | 9.76 | 5.47 | 2.78 | 41.71 |

Table 3: Results of training or freezing the visual encoder (VE) in LLaVA during preference optimization. × and ✓ denote the visual encoder states of training and freezing, respectively.

| Model | VE | MMHal | | AMBER | | | ObjHal | |
|---|---|---|---|---|---|---|---|---|
| | | Ova.↑ | R.↓ | CHAIR↓ | Cover↑ | Hal↓ | R.↓ | M.↓ |
| LLaVA | - | 2.75 | 42.7 | 8.30 | 61.0 | 48.6 | 14.1 | 7.4 |
| +DPO | × | **2.73** | **43.8** | 5.94 | 61.0 | 38.9 | 11.0 | 6.6 |
| +DPO | ✓ | 2.71 | 44.8 | **5.88** | **61.6** | **38.3** | 10.1 | **5.7** |
| +CHiP | × | **2.89** | **39.6** | 3.72 | 57.8 | 24.5 | **4.9** | **3.2** |
| +CHiP | ✓ | 2.68 | 43.8 | 3.74 | 54.9 | 22.1 | 5.3 | 3.3 |

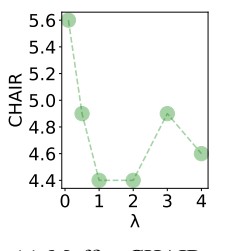
(a) Muffin+CHAIR

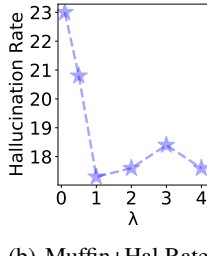
(b) Muffin+Hal Rate

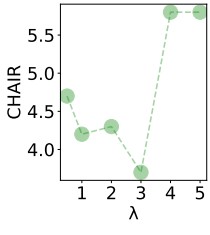
(c) LLaVA+CHAIR

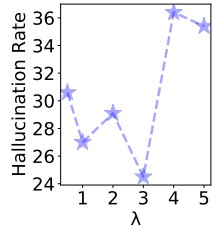
(d) LLaVA+Hal Rate

Figure 4: Results of Muffin+CHiP and LLaVA+CHiP evaluated on the AMBER dataset with different choices of weight $\lambda$ to control the strength of segment-level preference optimization. Findings: When $\lambda = 1$ ($\lambda = 3$), the best performance of the CHAIR and Hallucination Rate metric is achieved on AMBER based on Muffin (LLaVA-1.6).

**Impact of Training Paradigm.** The misalignment between image and text semantics is a significant cause of hallucinations in MLLMs (Liu et al., 2024a). However, most approaches (Wang et al., 2024) freeze the visual encoder and train only the connector and LLM during preference optimization. There raises a scientific question: Can full-model training during MLLM preference optimization reduce hallucinations? To investigate this, we explored the impact of freezing versus training the visual backbone on LLaVA enhanced by CHiP and DPO. The results are shown in Tab. 3. **Results:** DPO achieves a lower hallucination rate when the visual encoder is trained, whereas CHiP, which incorporates multi-level textual preference and visual preference optimization, does not achieve the expected reduction in hallucination rate when the visual encoder is trained. A possible reason is that the multiple optimization objectives may dilute the model's focus on image-text semantic alignment during the joint training of the visual encoder.

## 6 FURTHER ANALYSIS

### 6.1 GENERAL CAPABILITY ANALYSIS

Preference learning may compromise a model's general understanding capabilities. In this section, we evaluate and analyze the general capability performance of an MLLM enhanced by our CHiP. Specifically, we selected several popular general capability evaluation datasets, including MMMU (val) (Yue et al., 2024), MMMU (test), MMB-ENG (Liu et al., 2025), MMB-CN, ScienceQA (Lu et al., 2022), and LLaVA-Wild (Liu et al., 2024c). We compared the performance of LLaVA and LLaVA+CHiP on these datasets, with the results shown in Tab. 4.

**Observation**: We observe that LLaVA+CHiP outperforms LLaVA on five out of the six datasets. This indicates that CHiP slightly improves performance on the MMMU, LLaVA-Wild, and MMB-CN while maintaining comparable performance on others.

Table 4: The general capability evaluation results. Values in black indicate the best performance, in red show improvement with CHiP, and in green indicate a decline. Values with * are reproduced results. For LLaVA-Wild, we used *gpt-4o-2024-05-13* as evaluator due to *GPT-4-0314* was outdated; for MMMU-test, there was a lack of official LLaVA-1.6 reports.

|  | MMMU(val) | MMMU(test) | MMB-ENG | MMB-CN | ScienceQA | LLaVA-Wild |
|---|---|---|---|---|---|---|
| Num Samples | 900 | 10500 | 6666 | 6666 | 4241 | 90 |
| LLaVA | 35.80 | 31.70* | **67.40** | 60.60 | 70.10 | 74.90 |
| LLaVA+CHiP | **36.8**$^{+1.0}$ | **32.1**$^{+0.4}$ | 66.6$^{-0.8}$ | **60.82**$^{+0.22}$ | **70.15**$^{+0.05}$ | **76.2**$^{+1.3}$ |

## 6.2 IMPACT OF REJECTION IMAGE CONSTRUCTION STRATEGY

Preference sample quality depends on rejection image quality and its gap from the chosen image. In this section, we explore various strategies for constructing rejection images.

**Strategies.** The construction rejection images are listed below: (1) **Diffusion**: Following the forward diffusion process in image generation (Ho et al., 2020), small amounts of Gaussian noise are gradually added to the chosen image for T=500 steps. (2) **Blackness**: set all the RGB values of the chosen image to 0. (3) **Crop**: random cropping strategy is utilized to the chosen image. (4) **Rotation**: randomly rotate the chosen image by 10 to 80 degrees. (5) **Randomness**: select an image from the training set randomly.

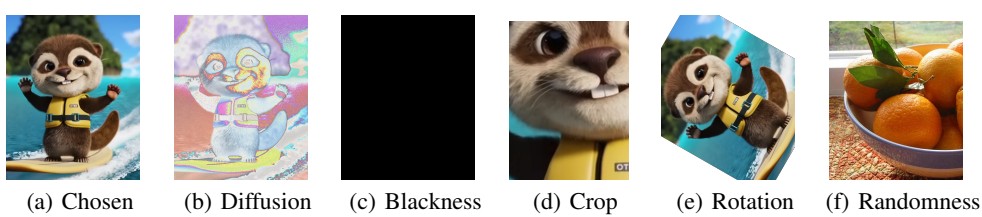

| (a) Chosen | (b) Diffusion | (c) Blackness | (d) Crop | (e) Rotation | (f) Randomness |

Figure 5: Examples of rejection images constructed by different strategies. (a) is the chosen image.

**Results.** The experimental results of CHiP under different construction strategies of rejection images are shown in Fig. 5. Observations: **The high similarity between the rejection and chosen images can lead to better preference optimization with CHiP.** The `diffusion` and `cropping` strategies represent blurred and sub-images of the chosen image, respectively, both retaining a significant amount of the chosen image's visual information, resulting in better performance. However, the `blackness` and `randomness` strategies by completely masking and replacing the chosen image

Table 5: Results of CHiP under different rejection image construction strategies. The **bold** values indicate the best performance. Observation: CHiP achieves the best performance with the diffusion strategy constructed rejection images.

| Strategy | ObjHal | | MMHal | |
|---|---|---|---|---|
| | R.↓ | M.↓ | Ova.↑ | R.↓ |
| Diffusion | **4.9** | **3.2** | **2.9** | **39.6** |
| Black | 9.4 | 5.0 | 2.4 | 43.8 |
| Cropping | 5.8 | 3.6 | 2.8 | 40.6 |
| Random | 10.9 | 5.9 | 2.9 | 41.7 |
| Rotate | 7.8 | 4.4 | 2.8 | 43.8 |

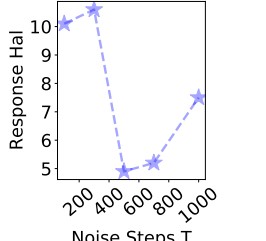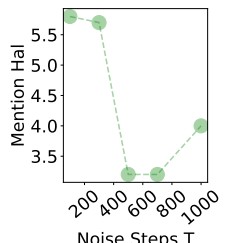

Figure 6: Results of LLaVA+CHiP evaluated on the ObjHal dataset with different values of noise step T. "Response" represents the response-level hallucination rate, while "Mention" represents the mention-level hallucination rate.

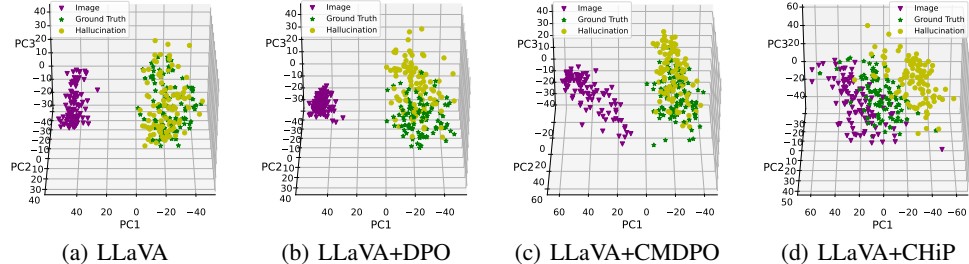

(a) LLaVA  (b) LLaVA+DPO  (c) LLaVA+CMDPO  (d) LLaVA+CHiP

Figure 7: The visualization of representation distributions of various preference optimization strategies. Findings: our CHiP makes more alignment between images and non-hallucination descriptions and improves the model's ability to distinguish between hallucinatory and non-hallucinatory text.

almost do not retain the information of the chosen image, leading to poorer performance. Although `rotation` preserves much of the chosen image's information, it creates significant differences in visual tokens after tokenization, resulting in poor performance.

**Impact of Noise Step T.**  We further explored the impact of noise steps T on the performance of CHiP, as shown in Fig. 6 for the ObjHal dataset. CHiP performs best at T=500. Possible reasons: (1) Fewer noise steps make the rejected image too similar to the chosen one, causing label ambiguity and weakening optimization. (2) Too many noise steps erase image details, making distinctions too easy and limiting the module's use of visual information.

### 6.3 REPRESENTATION VISUALIZATION

Ideally, a MLLM with a low hallucination rate should ensure that the representations of the textual description for the image are as close as possible to the representations of the image itself, while keeping the representations of hallucinated and non-hallucinated texts as far apart as possible. We analyze the effectiveness of CHiP compared to DPO and several ablation preference optimization strategies from a representational perspective. Specifically, we sampled 150 image-ground truth description pairs from the COCO-2017 (Lin et al., 2014) validation set, and GPT-4 was used to generate more detailed non-hallucinated descriptions (manually verified), as well as hallucinated descriptions. We then input the images, non-hallucinated texts, and hallucinated texts into LLaVA separately, and took the representation of the last token of the LLM (in this case, LLaMA) as the representation of the text or the image. We applied Principal Component Analysis (PCA) [2] to reduce the dimensionality of the high-dimensional representations, and the results are shown in Fig. 7.

**Observations:** Compared to LLaVA, DPO struggles with aligning image and text representations but can distinguish hallucinated from non-hallucinated text. After introducing image preference optimization based on DPO, namely CMDPO, the model not only distinguishes between hallucinated and non-hallucinated texts but also brings the representations of the image and the ground-truth description closer. With the introduction of more fine-grained text and image preference optimization, namely CHiP, the alignment between the image and ground-truth descriptions becomes even closer, while maintaining the ability to distinguish between hallucinated and non-hallucinated texts.

## 7 CONCLUSION

In this paper, we tackle the issue of hallucinations in multimodal large language models (MLLMs) by proposing Cross-Modal Hierarchical DPO (CHiP). CHiP integrates visual and hierarchical textual preference optimization, facilitating cross-modal preference capture and finer-grained distinctions. Experiments on four widely-used datasets demonstrate that CHiP effectively reduces hallucinations. We visualized the representations of images, non-hallucinatory descriptions, and hallucinatory descriptions. The results show that CHiP, compared to standard multimodal DPO, more effectively bridges the semantic gap between images and non-hallucinatory descriptions while enhancing the distinction between hallucinatory and non-hallucinatory descriptions.

---

[2]https://en.wikipedia.org/wiki/Principal_component_analysis

ACKNOWLEDGMENTS

This research is supported by A*STAR, CISCO Systems (USA) Pte. Ltd and National University of Singapore under its Cisco-NUS Accelerated Digital Economy Corporate Laboratory (Award I21001E0002). This work was also supported by the National Key Research and Development Program of China (No. U24B20181).

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

# A  DIFFERENCES BETWEEN TOKEN- AND RESPONSE-LEVEL OPTIMIZATION

Here, we first summarize the differences between token-level optimization and response-level optimization (in Sec. A.1). For clarity, the following two subsections provide detailed derivations of response-level (in Sec. A.2) and token-level (in Sec. A.3) optimization for reference.

## A.1  OVERVIEW

The preference optimization function can be divided into two components: (1) the reward function, which quantifies user preferences and drives the optimization direction; and (2) KL divergence, which controls the difference between the output distributions of the policy model and the preference model.

Given the input instruction $x$, image $m$, and response $y$, the **Response-level Preference Optimization** can be formulated as below:

$$\max_{\pi_\theta} \mathbb{E}_{x,m\sim\mathcal{D},y\sim\pi_\theta(y|x)}\big[r(x,m,y)\big] - \beta\mathbb{D}_{\mathrm{KL}}\big[\pi_\theta(y \mid x,m) \mid\mid \pi_{\mathrm{ref}}(y \mid x,m)\big], \tag{13}$$

where the reward function can be denoted as:

$$r(x,m,y) = \beta\log\frac{\pi_\theta(y|x,m)}{\pi_{\mathrm{ref}}(y|x,m)} + \beta\log Z(x,m)\,, \tag{14}$$

And, the **Token-level Preference Optimization** can be defined as below:

$$\max_{\pi_\theta} \mathbb{E}_{x,m,y^{<t}\sim\mathcal{D},y^t\sim\pi_\theta(\cdot|[x,m,y^{<t}])}\big[r(x,m,y)\big] - D_{\mathrm{SeqKL}}(x,m,y;\pi_\theta\|\pi_{ref})), \tag{15}$$

where the reward function can be denoted as:

$$r(x,m,y) = \sum_{t=1}^{T}\gamma^{t-1}R([x,m,y^{<t}],y^t) \tag{16}$$

where $D_{\mathrm{SeqKL}}(x,m,y;\pi_\theta\|\pi_{ref})$ denotes the sequential KL divergence, and it can be defined as:

$$D_{\mathrm{SeqKL}}(x,m,y;\pi_\theta\|\pi_{ref}) = \sum_{t=1}^{T} D_{\mathrm{KL}}(\pi_\theta(\cdot|[x,m,y^{<t}])\|\pi_{ref}(\cdot|[x,m,y^{<t}])). \tag{17}$$

**Comparison:**

**(1) Reward Function**   For response-level preference optimization, the reward function is calculated based on the generation probabilities of a response by the policy model and the reference model (as shown in Eq. 14). The probability of the entire response is first obtained, and then the reward for the entire response is calculated.

For token-level preference optimization, the reward function calculates the reward for each token individually (for example, $y^t$, based on the $x$ and $m$, and $y^{<t}$) and then sums up the rewards of all tokens to obtain the reward for the entire response, as shown in Eq. 16.

**(2) KL Divergence**   For response-level optimization (as shown in Eq. 13), KL divergence is calculated as the distance between the distributions of the response $y$ given $x$ and $m$, as modeled by the policy model and the reference model (note: this is based on the response distribution).

For token-level optimization, KL divergence is computed as the distance between the distributions of $y^t$ given $x$ and $m$, and $y^{<t}$, as modeled by the policy model and the reference model. The overall distance between the policy and reference models is obtained by summing these distances across all tokens. Eq. 17 formulate this process.

## A.2  RESPONSE-LEVEL PREFERENCE OPTIMIZATION

Direct Preference Optimization (DPO) (Rafailov et al., 2024) is primarily a preference optimization method that focuses on aligning language models with human preferences without the need for

explicit reward modeling or reinforcement learning. Given a model to be optimized $\pi_\theta$, and the reference policy $\pi_{\text{ref}}$, which is a supervised fine-tuning model, the RL optimization of RLHF can be formulated as:

$$\max_{\pi_\theta} \mathbb{E}_{x,m \sim \mathcal{D}, y \sim \pi_\theta(y|x)} \big[ r(x,m,y) \big] - \beta \mathbb{D}_{\text{KL}} \big[ \pi_\theta(y \mid x,m) \mid\mid \pi_{\text{ref}}(y \mid x,m) \big]. \tag{18}$$

By maximizing the KL-constrained reward objective to obtain the optimal solution and establishing a mapping between the reward model and the optimal policy, the representation of the reward function is derived as follows:

$$r(x,m,y) = \beta \log \frac{\pi_\theta(y|x,m)}{\pi_{\text{ref}}(y|x,m)} + \beta \log Z(x,m), \tag{19}$$

where $x$ is the input instruction, $m$ is the image, $y$ is the response, $\beta$ is a constant, and $Z(x,m)$ is the partition function.

Given the chosen response $y_w$, where the evaluator preferred it over the rejected response $y_l$, DPO is expected to learn to maximize the reward difference between chosen $(y_w)$ and rejected responses $(y_l)$. The preference optimization objective becomes:

$$\begin{aligned} \mathcal{L}_{\mathcal{DPO}} &= -\mathbb{E}_{(x,m,y_w,y_l)} \big[ \log \sigma(r(x,m,y_w) - r(x,m,y_l)) \big] \\ &= -\mathbb{E}_{(x,m,y_w,y_l)} \big[ \log \sigma(\beta \log \frac{\pi_\theta(y_w|x,m)}{\pi_{\text{ref}}(y_w|x,m)} - \beta \log \frac{\pi_\theta(y_l|x,m)}{\pi_{\text{ref}}(y_l|x,m)}) \big], \end{aligned} \tag{20}$$

where DPO learns preferences based on the ranking of the entire response, and the action score can be formulated as:

$$\log \pi(y|x,m) = \sum_{y_i \in y} \log p(y_i|x,m,y_{<i}), \tag{21}$$

where $y_i$ denotes the $i$-th token of the response $y$. During DPO training, the reference model $\pi_{\text{ref}}(y|x,m)$ is usually kept fixed while the policy model $\pi_\theta(y|x,m)$ is updated.

## A.3 TOKEN-LEVEL PREFERENCE OPTIMIZATION

The objective function of DPO operates at the sentence level, as shown in Eq. 18. The principle of token-level preference optimization is similar to sentence-level preference optimization. The difference between them lies in the reward function. In token-level preference optimization, the reward function is token-wise, which can be viewed as the cumulative reward for generating the text.

Given a response composed of $T$ tokens $y = [y^1, y^2, ..., y^T]$, where $y^t \in \mathcal{Y}$, and $\mathcal{Y}$ represents the vocabulary. Additionally, we define $y^{<1} = [\,]$. Given a prompt $x$, a image $m$, and model-generated response $y$'s first $t-1$ tokens, the LM predicts the probability distribution of the next token can be formulated as $\pi_\theta(\cdot|[x,m,y^{<t}])$. Therefore, the objective of token-level preference optimization can be denoted as below:

$$\max_{\pi_\theta} \mathbb{E}_{x,m,y^{<t} \sim \mathcal{D}, y^t \sim \pi_\theta(\cdot|[x,m,y^{<t}])} \big[ \sum_{t=1}^{T} \gamma^{t-1} R([x,m,y^{<t}], y^t) \big] - D_{\text{SeqKL}}(x,m,y; \pi_\theta \| \pi_{ref})), \tag{22}$$

where $D_{\text{SeqKL}}(x,m,y; \pi_\theta \| \pi_{ref})$ denotes the sequential KL divergence, and it can be defined as:

$$D_{\text{SeqKL}}(x,m,y; \pi_\theta \| \pi_{ref}) = \sum_{t=1}^{T} D_{\text{KL}}(\pi_\theta(\cdot|[x,m,y^{<t}]) \| \pi_{ref}(\cdot|[x,m,y^{<t}])). \tag{23}$$

where $\sum_{t=1}^{T} \gamma^{t-1} R([x,m,y^{<t}], y^t)$ is the accumulate reward, and $\gamma$ represents a weight and is a constant.

We can follow a similar approach to response-level preference optimization by maximizing the KL-constrained reward objective to obtain the optimal solution, thereby deriving a reward function similar to Eq. 19. And further derive the token-level preference optimization function like sentence-level

shown in Eq. 20. However, $Z([x, m, y_w^{<t}]; \beta) \neq Z([x, m, y_l^{<t}]; \beta)$, which means that optimization at the sentence-level preference pairs can result in the cancellation of policies, while the cancellation does not occur in token-level preference optimization.

Here, we directly employ the Bradley-Terry model to represent the probability of human preferences based on the optimal policy. In the KL-constrained advantage maximization problem associated with Eq. 22, the Bradley-Terry model takes the optimal policy $\pi_\theta$ and the reference policy $\pi_{\text{ref}}$ to expresses human preference probabilities:

$$P_{\text{BT}}(y_w \succ y_l | x, m) = \sigma(\lambda(x, m, y_w, y_l) - \delta(x, m, y_w, y_l)), \tag{24}$$

where, $\lambda(x, m, y_w, y_l)$ refers to the difference in rewards implicitly defined by the language model $\pi_\theta$ and the reference model $\pi_{\text{ref}}$ (Rafailov et al., 2024), represented as

$$\lambda(x, m, y_w, y_l) = \beta \log \frac{\pi_\theta(y_w \mid x, m)}{\pi_{\text{ref}}(y_w \mid x, m)} - \beta \log \frac{\pi_\theta(y_l \mid x, m)}{\pi_{\text{ref}}(y_l \mid x, m)}, \tag{25}$$

and $\delta(x, m, y_w, y_l)$ is the difference in sequential forward KL divergence between the preference pairs $(x, m, y_w)$ and $(x, m, y_l)$.

$$\delta(x, m, y_w, y_l) = \beta D_{\text{SeqKL}}(x, m, y_w; \pi_{\text{ref}} \| \pi_\theta) - \beta D_{\text{SeqKL}}(x, m, y_l; \pi_{\text{ref}} \| \pi_\theta), \tag{26}$$

where $\beta$ is the weight. Put them together, we obtain the loss function for token-level preference optimization:

$$
\begin{aligned}
\mathcal{L}_{\text{TDPO}} &= -\mathbb{E}_{(x,m,y_w,y_l)} \left[ \log \sigma \left( \lambda(x, m, y_w, y_l) - \delta(x, m, y_w, y_l) \right) \right], \\
&= -\mathbb{E}_{(x,m,y_w,y_l)} \left[ \log \sigma \left( \beta \log \frac{\pi_\theta(y_w \mid x, m)}{\pi_{\text{ref}}(y_w \mid x, m)} - \beta \log \frac{\pi_\theta(y_l \mid x, m)}{\pi_{\text{ref}}(y_l \mid x, m)} \right. \right. \\
&\quad \left. \left. - \alpha \left( \beta D_{\text{SeqKL}}(x, m, y_2; \pi_{\text{ref}} \| \pi_\theta) - sg \left( \beta D_{\text{SeqKL}}(x, m, y_1; \pi_{\text{ref}} \| \pi_\theta) \right) \right) \right) \right],
\end{aligned}
\tag{27}
$$

**Token-level Preference Optimization ($\mathcal{L}_{\mathcal{PO}k}$).** The difference in reward $\lambda(x, m, y_w, y_l)$ appears in both the response-level and segment-level preference optimizations in this work. Therefore, we consider solely on sequential KL divergence, which serves as the optimization term for token-level preference optimization:

$$\mathcal{L}_{\mathcal{PO}k} = sg \left( \beta D_{\text{SeqKL}}(x, m, y_w; \pi_{\text{ref}} \| \pi_\theta) \right) - \beta D_{\text{SeqKL}}(x, m, y_l; \pi_{\text{ref}} \| \pi_\theta), \tag{28}$$

where $sg$ represents the stop-gradient operator, and

$$D_{\text{SeqKL}}(x, m, y; \pi_{\text{ref}} \| \pi_\theta) = \sum_{t=1}^{T} D_{\text{KL}}(\pi_{\text{ref}}(y|x, m, y^{<t}) \| \pi_\theta(y|x, m, y^{<t})). \tag{29}$$

# B   FURTHER ANALYSIS

## B.1   DOES CHiP MAKE THE MODEL LESS TALKATIVE?

**Why does CHiP lower performance on the metric of `Cover` (Object coverage of responses)?** The results in Tab. 1 show that while CHiP reduces the hallucination rate, it also decreases object coverage on the AMBER dataset. This raises a question: does CHiP reduce hallucinations by limiting the amount of text generated? To answer this question, we calculated the average output length of LLaVA, LLaVA+DPO, and LLaVA+CHiP across three generative datasets, namely AMBER, MMHal, and ObjHal. Since HallusionBench is a multiple-choice task, calculating response length is not meaningful, so we have omitted it here. The results are presented in Tab. 6. We can observe that the output lengths of the three models are comparable. Therefore, CHiP lowers the hallucination rate without making the model less talkative. Furthermore, a manual analysis of LLaVA+CHiP's responses reveals that when an image contains ambiguous or uncertain objects or attributes, CHiP tends to omit to mention them, effectively reducing hallucinations.

**Why does CHiP lower performance in `fA` (Figure Accuracy)?**

The `fA` metric is designed to evaluate the model's logical consistency, specifically ensuring that responses to questions are not based on random guesses. From Tab. 1, we observe that on LLaVA-7B, CHiP achieves a lower `fA` compared to DPO, but higher than the original LLaVA-7B. A possible reason is that DPO directly targets response-level preference optimization, focusing on aligning the model's outputs with human preferences at the response level, which makes it better at maintaining logical consistency. In contrast, CHiP's optimization objectives include segment-level, token-level, and image preference optimization. This additional complexity may dilute the focus on logical consistency during the optimization process, resulting in CHiP's slightly lower `fA` compared to DPO.

Table 6: Average output length statistics of different models.

| Model | AMBER | MMHal | ObjHal |
|---|---|---|---|
| LLaVA | 131 | 44 | 164 |
| LLaVA+DPO | 127 | 40 | 160 |
| LLaVA+CHiP | 124 | 43 | 151 |

## B.2 ANNOTATOR BACKGROUND IN HUMAN EVALUATION

To mitigate the errors in GPT evaluations caused by incomplete annotations in the MMHal dataset, we introduced human experts for manual evaluation to ensure more reliable conclusions: CHiP has a lower hallucination rate compared to traditional DPO. Details about the annotators: We invited three human experts who specialize in multimodal hallucination tasks. These experts are well-versed in various types of hallucinations, such as object hallucinations, attribute hallucinations, and environment hallucinations. We consider object hallucinations to be more severe than attribute and environment hallucinations. This prioritization allows the expert to efficiently classify and label hallucination types during the annotation process, enabling them to assign reasonable and accurate scores.

## B.3 EFFECT OF TOKEN-LEVEL PREFERENCE OPTIMIZATION

In this section, we explored whether using token-level preference optimization independently as an optimization objective would still allow the model to work effectively. The evaluation results on four datasets are presented in the Tab. 7 It can be observed that solely using token-level preference optimization results in lower hallucination rates compared to traditional DPO and the original LLaVA, which holds for the four datasets.

Table 7: Performance comparison of LLaVA after token-level preference optimization (TDPO) and direct preference optimization (DPO). Values in bold indicate the best performance.

| Model | ObjHal | | MMHal | | HallusionBench | | | AMBER | | | |
|---|---|---|---|---|---|---|---|---|---|---|---|
| | Resp.↓ | Mention.↓ | Overall↑ | Resp.↓ | (qAcc)↑ | (fAcc)↑ | (aAcc)↑ | CHAIR↓ | Cover↑ | Hal↓ | Cog↓ |
| TDPO | **9.56** | **5.50** | **2.73** | **42.71** | **22.64** | **28.32** | **57.22** | **5.90** | **61.30** | **37.60** | 3.10 |
| DPO | 11.03 | 6.61 | 2.73 | 43.75 | 22.20 | 28.32 | 56.60 | 5.90 | 61.00 | 38.90 | **3.00** |

## B.4 FINE-GRAINED ANALYSIS

Here, we conducted a fine-grained evaluation of the base model (i.e., Muffin and LLaVA), DPO, and CHiP on the **AMBER**, **MMHal-Bench (MMHal)**, and **Object HalBench (ObjHal)** datasets. The results are shown in Tab. 8, Tab. 9, and Tab. 10. The main finding is that CHiP performed well on many fine-grained evaluation metrics across these three datasets.

Table 8: Fine-grained results on **AMBER**. Bold values indicates the best performance. ↑ indicates that a higher value represents better performance, while ↓ indicates that a lower value is better. Findings: Our CHiP achieves the best `AMBER Score` under both the base model Muffin and the LLaVA model.

| Model | Generative | | | | Discriminative | | | | AMBER |
|---|---|---|---|---|---|---|---|---|---|
| | CHAIR↓ | Cover↑ | Hal↓ | Cog↓ | F1↑ | F1E↑ | F1A↑ | F1R↑ | Score↑ |
| Muffin (13B) | 8.0 | **48.3** | 32.1 | 3.5 | 86.4 | 95.0 | 79.3 | 71.4 | 89.20 |
| +DPO | 6.2 | 46.9 | 26.5 | 2.5 | 86.9 | 95.9 | 79.9 | 70.4 | 90.35 |
| +CHiP | **4.4** | 45.3 | **17.6** | **1.5** | **87.6** | **96.1** | **80.5** | **73.3** | **91.60** |
| LLaVA-1.6 (7B) | 8.3 | **61.0** | 48.6 | 4.2 | 87.0 | 95.1 | **81.5** | **69.6** | 89.35 |
| +DPO | 5.9 | **61.0** | 38.9 | 3.0 | **87.4** | 97.8 | 81.3 | 63.4 | 90.75 |
| +CHiP | **3.7** | 57.8 | **24.5** | **1.6** | 86.9 | **98.3** | 80.3 | 62.0 | **91.60** |

Table 9: Fine-grained results on **MMHal-Bench (MMHal)**. Bold values indicates the best performance. ↑ indicates that a higher value represents better performance, while ↓ indicates that a lower value is better. Findings: Our CHiP achieves the best `Overall score` and `Hallucination rate` under both the base model Muffin and the LLaVA model.

| Model | Overall↑ | Hallu↓ | Score in Each Question Type ↑ | | | | | | | |
|---|---|---|---|---|---|---|---|---|---|---|
| | | | Attribute | Adversarial | Comparison | Counting | Relation | Environment | Holistic | Other |
| Muffin | 2.41 | 60.42 | **2.67** | **3.17** | 2.83 | 2.83 | 2.42 | **3.00** | 2.08 | 0.25 |
| +DPO | 2.49 | 52.08 | 3.50 | 2.33 | **2.92** | 2.08 | 2.50 | 2.67 | **2.33** | 1.58 |
| +CHiP | **2.58** | **48.96** | 3.58 | 2.58 | 2.08 | **3.42** | **2.83** | 2.58 | 1.50 | **2.08** |
| LLaVA | 2.78 | 42.71 | 3.75 | **3.50** | **3.50** | 1.50 | 1.92 | 4.08 | **1.75** | **2.83** |
| +DPO | 2.73 | 43.75 | **4.17** | 2.92 | 3.00 | **2.67** | **2.67** | **4.25** | 1.17 | 1.00 |
| +CHiP | **2.84** | **39.58** | **4.17** | 3.33 | 2.67 | **2.67** | 2.25 | 4.08 | 1.67 | 1.92 |

Table 10: Fine-grained results on **Object HalBench (ObjHal)**. Bold values indicates the best performance. ↑ indicates that a higher value represents better performance, while ↓ indicates that a lower value is better. Findings: Our CHiP performs best on all the fine-grained evaluation metrics (except for the Object Recall) under the base model Muffin and the LLaVA model.

| Model | Response Hall↑ | Object Hall↑ | Response Correct↑ | Object Correct↑ | Object Recall↑ |
|---|---|---|---|---|---|
| Muffin | 21.53 | 11.61 | 78.47 | 88.39 | **56.29** |
| +DPO | 10.65 | 5.18 | 89.35 | 94.82 | 51.78 |
| +CHiP | **6.17** | **3.91** | **93.83** | **96.09** | 40.99 |
| LLaVA | 14.08 | 7.37 | 85.92 | 92.63 | **55.03** |
| +DPO | 11.03 | 6.61 | 88.97 | 93.39 | 52.83 |
| +CHiP | **4.92** | **3.21** | **95.08** | **96.79** | 48.95 |

---

Your task consists of two parts: one is to provide a more detailed and accurate description of no less than 100 wordsbased on the ground truth caption of the image. Secondly, modify the description you generated in the first step to create a similar but incorporate many elements of hallucination.

Here are three examples for your reference:
–Start of examples–
Example-1:
Ground Truth Caption:
A white plate with a sandwich and sides on it.
Non-Hallucinative Caption:
A white plate with a sandwich and sides on it. The sandwich is made of two slices of bread with a filling inside. The sides are a small bowl of white creamy dip and some fried onions. There is also a side dish of sliced oranges in a clear glass bowl.
Hallucinative Caption:
A white plate with a sandwich and sides on it. The sandwich is made of two slices of bread with a filling of ham and cheese inside. The sides are a small bowl of creamy dill dip and some crunchy onions. There is also a side dish of sliced oranges in a clear glass bowl, served with a glass of wine. The plate is placed on a table with a placemat made of woven reeds.
Example-2:
Ground Truth Caption:
An orange truck driving down a street full of men in the back.
Non-Hallucinative Caption:
An orange dump truck is driving down the street with several men standing in the back of the truck bed. All the men are wearing red caps and white shirts. There is a smaller yellow bus to the left of the truck, and there is a light pole in the middle of the frame.
Hallucinative Caption:
An orange dump truck is driving down a street full of people in the back, all of them wearing red caps and white shirts. The truck is driving past a yellow bus that is full of people and is about to pass a fire hydrant on the side of the road. The truck is driving very fast and there are police cars chasing it. The men in the back of the truck are holding their hands in the air and yelling, "We're not gonna stop! We're gonna take back the streets!".
Example-3:
Ground Truth Caption:
A pizza sitting on top of a pizza box covered in cheese.
Non-Hallucinative Caption:
A pizza with cheese, tomato sauce and toppings sits inside a cardboard pizza box. The pizza appears to be cut into 6 slices and the box has some grease stains.
Hallucinative Caption:
A perfectly cooked pizza sits inside a cardboard pizza box. The pizza is covered in melted cheese and pepperonis and there are traces of red pepper flakes around the edges. A small, green pepper rests on top of the pizza. The box is stained with a sauce that is most likely marinara.
–End of examples–

Ground Truth Caption:
{caption}
Please output in this format:
[Non-Hallucinative Caption:]
[Hallucinative Caption:]

---

Table 11: The prompt used to generate hallucinated and non-hallucinated descriptions for a given input image.

## B.5 PROMPT

To reveal the limitations of existing multimodal DPO from the representation perspective and demonstrate how our method, CHiP, effectively mitigates these limitations, we extracted 150 images and their ground-truth descriptions from the COCO-2017 (Lin et al., 2014) validation set. Gemini (The version we choose to use is gemini-1.5-flash-latest) was then used to generate more detailed non-hallucinated and hallucinated descriptions, with the non-hallucinated descriptions manually verified. Tab. 11 presents the prompts we used to guide Gemini in generating these detailed non-hallucinated and hallucinated descriptions.

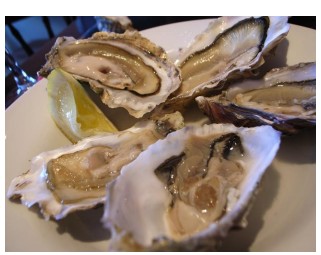

**LLaVA**

There are six oysters visible in the photo.

**LLaVA + DPO**

There are six oysters in the photo.

**LLaVA + CHiP**

There are 5 oysters in the photo.

**Question:** How many oysters can you see in the photo?

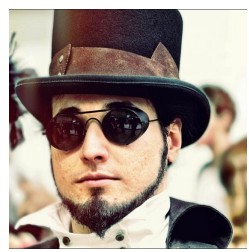

**LLaVA**

In the reflection from the sunglasses, you can see a blurred image of what appears to be a person wearing a top hat and sunglasses. The person in the reflection seems to be wearing a similar outfit to the person in the foreground, which includes a top hat and sunglasses, but the details are not clear due to the blur.

**LLaVA + DPO**

In the reflection from the sunglasses, you can see the person wearing the sunglasses and the person in the background. The person in the background appears to be wearing a white shirt.

**LLaVA + CHiP**

In the reflection from the sunglasses, you can see what appears to be the upper part of another person's head and shoulders, suggesting that the person wearing the sunglasses is not alone in the image. The reflection is somewhat blurred, indicating that the sunglasses may have a reflective surface that distorts the image being reflected. The details of the other person's appearance are not clear due to the reflection's distortion.

**Question:** What can you see in the reflection from the sunglasses?

Figure 8: Qualitative results of LLaVA+CHiP compared with LLaVA+DPO and LLaVA on MMHal-Bench dataset. Correct answers and hallucinations are highlighted in green and red, respectively.

## C  QUALITATIVE RESULT

To demonstrate the effectiveness of our approach, we present qualitative results on the MMHal-Bench dataset in this section, as shown in Fig. 8. LLaVA+CHiP demonstrates a significant reduction in hallucinations in text generation tasks. This improvement can be attributed to the multi-level preference optimization, which enables the model to capture image-text relationships across varying granularities. Additionally, the visual preference optimization module enhances semantic alignment between the two modalities, further contributing to the reduction in hallucinations.

