# OpenReview forum: "CHiP: Cross-modal Hierarchical Direct Preference Optimization for Multimodal LLMs"
_ICLR.cc/2025/Conference — ICLR 2025 Poster_

### Official Review · Reviewer_xtwk · 2024-11-01

**Soundness:** 2
**Presentation:** 3
**Contribution:** 2
**Rating:** 5
**Confidence:** 3

**Summary:**

Direct Preference Optimization (DPO) is frequently used to address the hallucination problem in Vision-Language Models (VLMs). The paper identifies a limitation with DPO, noting that it does not effectively align image and text representations, and proposes Cross-modal Hierarchical Direct Preference Optimization (CHiP) as a solution to this issue.

**Strengths:**

- The paper is highly readable.
- It presents a novel perspective on the limitations of traditional DPO, highlighting that it does not apply weight for important segments, resulting in the possibility of applying DPO even to non-hallucinatory segments of a rejected response.
- The idea of assigning higher rewards to important segments is innovative and convincing.
- The approach of learning both Textual and Visual Preferences is new and valuable.
- The experiments effectively demonstrate the hallucination mitigation, and the overall results are strong.

**Weaknesses:**

- While the paper mentions that Visual Preference Optimization trains the model to understand images that better match the chosen response, the concept of "Visual Preference" itself is unclear.
- The use of augmented images as rejection images is confusing, as it is unclear why semantically similar images are used as rejection images. In Figure 5, except for (c) Blackness and (f) Random, the other images share the same semantics. This approach differs from typical augmentation techniques that increase model robustness by training with transformed images. If (c) and (f) are used as rejection images, it can be understood as a way to prevent generating responses without observing the image. However, other images retain the same semantics, making it difficult to consider them as proper rejection images.

**Questions:**

- How are segments determined in Segment-level Preference Optimization? Are partially pre-modified sentences prepared for this purpose?
- I initially thought Response-level Preference Optimization also involved optimization at the token level. Could you clarify the difference between Token-level and Response-level Preference Optimization?
- In Equation (8), does "sg" refer to stop gradient? I’m curious about the reasoning behind this.
- What could be the reason for the lower performance in fA and Cover in Table 1?
- In Table 3, what insight is expected from the experiment comparing training vs. freezing the Visual Encoder, and what is the purpose of this comparison?

---

### Official Review · Reviewer_Gv4h · 2024-11-01

**Soundness:** 3
**Presentation:** 2
**Contribution:** 3
**Rating:** 6
**Confidence:** 4

**Summary:**

This paper introduces Cross-modal Hierarchical Direct Preference Optimization (CHiP). This approach incorporates a visual preference optimization module alongside a hierarchical textual preference optimization module (preference learning in response, segment, and token-level), allowing MLLMs to learn from both visual and textual preferences across various levels of granularity.

**Strengths:**

1. This paper proposes to eliminate hallucinations from multiple levels, including response-level, segment-level, and token-level.

2. The proposed Visual Preference Optimization is interesting, make model less over-relied on language counter part.

3. The proposed method achieves solid results on multiple hallucination benchmarks.

**Weaknesses:**

1. The technical details of some methods are not clear enough.

2. lacks experiments on general capability benchmarks.

3. The selection of rejection image in proposed Visual Preference Optimization remain improvement.

**Questions:**

1. The explanation of Segment-level Preference Optimization is not clear. Author claims that "we assign higher rewards to the segments that differ between the chosen response and the rejection response.". How the non-hallucinated and hallucinated segment pair is determined? If not human labeled, will the wrongly labeled segment affect optimization effect? Meanwhile, the Token-level Preference Optimization seems strange. From the equation9 we can see the whole sequence is calculated during the reward calculating, but why use the whole sentence when non-hallucinated and hallucinated segment pair is labeled? Will the token-level strategy work if only consider the token in labled segment?

2. Lacks general capability evaluation, it is known that preference learning may harm general understanding capability. Can arthors provide results on general capability benchmarks, such as MMMU and MMBench? Meanwhile, more human evaluation is required, does proposed method eleiminate hallucination by making model less talkative, for example, the cover rate in AMBER decreases after optimization.

3. The proposed visual preference optimization and the result is interesting. However the IMAGE CONSTRUCTION STRATEGY remain exploration. From results, we can see that rejection image that differs a lot from original image can lead to sub-optimal results. A strategy may outperform diffusion is finding rejection images as close as possible. For example, in figure5, a rejection image maybe also include a bear, ocean, mountain, but inconsistent with the correct response.

4. Formatting errors. Name unclear and inconsistence (CMDPO in Table3 but not mentioned, and HCMDPO in figure9), table 6 written as figure.

---

### Official Review · Reviewer_NXL3 · 2024-11-01

**Soundness:** 3
**Presentation:** 4
**Contribution:** 2
**Rating:** 8
**Confidence:** 4

**Summary:**

Introduces a new optimisation recipe to reduce hallucinations for multimodal models.
Hierarchical Textual Preference Optimisation tries to solve the problem that response-level preference optimizations have: they can’t clearly identify what segments/tokens contain hallucinations. They consider three levels of preference optimisation for text: response-level, segment-level, and token-level.
Visual Preference Optimisation tries to reduce reliance on large language models, which start hallucinating (ignoring the image) once the image is not well aligned. They take the ground-truth image and create a synthetic rejection image (using e.g. rotating, cropping, adding noise etc).
Combining both the hierarchical textual preference optimisation, on all three levels, and the visual preference optimisation, they introduce CHiP. They ablate each component individually, but show that combining all significantly improves performance on hallucination benchmarks.

(update: and they show equality in non-hallucination benchmarks.)

**Strengths:**

* All code and datasets are public.
* Well written.
* Is very clear about how baselines should be compared (not everything can be directly compared, but they still include them).
* Introduces all benchmarks and evaluation tasks/metrics in the paper with a quick overview.
* Results seem strong for the benchmarks this paper focuses on. Good analysis of the components that were used to achieve this.

**Weaknesses:**

1. While showing improvements on the benchmarks that this paper focused on, it does not show the impact on non-hallucination benchmarks. This makes it harder to estimate whether this is useful in training recipes in ‘the wild’.
2. All of the ablations/experiments to hill-climb results were done using the final evaluation sets (?). HallusionBench is used in the final overview, and it still outperforms DPO, but the results don’t seem to be as clear as the benchmarks that were mainly used during hyper parameter estimations.
3. Nit: "Reference results (these results cannot be directly comparable)" (page 7) -> "cannot be directly compared" or "are not directly comparable"
4. Nit: not sure if we already need to mention "We select the strategy with the best performance, which generates the rejection image by adding noise ... " (page 5) there, since it seems to be part of results.

**Questions:**

1. “Empirically, we set the weight of response-level preference optimisation to 1” (page 8) — can you share those empirical results?
2. “We found that the best performance was achieved when λ = 1 and λ = 3 for the Muffin and LLaVA-1.6 frameworks” (page 8) — are you not afraid of overfitting on the dataset with these hyperparamters? You’re not using a validation set vs test set here, right?
3. On page 6, "Implementation Details", how were 0.5 and 0.1 chosen?
3. What is the impact of this preference optimisation stage for the performance on existing benchmarks? You show it improves on hallucination benchmarks, does it keep its performance on previously measured benchmarks?
4. "To make the results more reliable, we invited experts to manually annotate the data to compare" (page 7) -- Do you have details about these annotators? How many/anything special that can be shared?

---

### Meta-Review · Area_Chair_T1WE · 2024-12-19

**Metareview:**

The paper studies hallucination in the MLLM. The author found that directly using DPO is not enough for alleviating the hallucination in the MLLM, as the model cannot distinguish the distribution b/w the non-hallucinated one and the hallucinated one. The author proposed a cross-modal hierarchical DPO to alleviate this issue. Concretely, the author proposed a combination loss on the response level, segment level, and token level to assign rewards at different granularity. The author also proposed to incorporate a preference loss on the image side.

Strength:
1. The paper is easy to read and easy to follow
2. The baselines are clearly presented
3. The proposed approach achieves good performance by reducing the hallucination.
4. The proposed approach is novel.

Weakness:
1. Didn't clearly describe the tech detail in the original version.
2. Didn't show the performance on non-hallucination benchmarks.

The author addressed those weaknesses during the rebuttal. I would recommend accept.

**Additional Comments On Reviewer Discussion:**

Reviewers are mainly complained about two things:
1. tech details are not very clear.
2. lack the non-hallucination benchmarks.
3. the selection of images used in the rejected samples.

During the rebuttal, the author responded both clearly. One reviewer improved the score. The second reviewer was satisfied with the results.

Only one reviewer voted borderline reject. During the rebuttal, the reviewer is satisfied with the response. Therefore, I would consider the concerns have been addressed.

---

### Decision · Program_Chairs · 2025-01-22

Accept (Poster)